# A Qualitative Examination of Water Access and Related Coping Behaviors to Understand Its Link to Food Insecurity among Rural Households in the West Region in Cameroon

**DOI:** 10.3390/ijerph17134848

**Published:** 2020-07-06

**Authors:** Carole D. Nounkeu, Jigna M. Dharod

**Affiliations:** Department of Nutrition, University of North Carolina at Greensboro, Greensboro, NC 27402-6170, USA; carolenounkeu@gmail.com

**Keywords:** rural areas, food insecurity, coping behaviors, water insecurity, water use, water quality

## Abstract

Food insecurity is a significant public health issue, since it causes malnutrition and engenders millions of deaths every year. A significant association is found between water and food insecurity. However, it remains unclear what are the pathways through which water shortage impacts food insecurity. Hence, a qualitative study was conducted in rural areas in Cameroon to (1) examine water access, its management, and its daily use and (2) investigate common behavior changes and coping strategies adults used in managing limited water availability in their households. Three rounds of focus group discussions and six key informant interviews were conducted with men and women. The results demonstrated that water access was limited, involving long walking distances and making several trips to the water sources. The household size, number of adults vs. children, and presence of storage containers affected water availability and its daily use. To manage limited water, coping behaviors included skipping drinking, changing cooking plans, and recycling water. In conclusion, limited water access increases food insecurity through several pathways. Governments, policy makers, and international organizations should recognize the interwoven link between water and food security. Joint actions and collaborative efforts are needed to improve success and reduce tradeoffs in achieving Sustainable Development Goals # 2 and # 6.

## 1. Introduction

Approximately 237 million people in sub-Saharan Africa are food insecure and experiencing chronic undernutrition due to it [1]. Specifically, West and Central Africa experience the highest level of food insecurity and hunger in sub-Saharan Africa [1]. Due to food insecurity, the prevalence of undernourishment in the form of inadequate weight gain during pregnancy and stunted growth among children has increased significantly in the region [1,2]. In 2019, 144 million children under five were stunted and approximately 47 million had low weight for their heights [2]. Food insecurity, defined as the inconsistent access to safe and nutritious food for an active and healthy life, has been associated with limited food intake, poor diet quality, and high rates of infectious diseases [3]. In estimating potential causes, weak commodity prices and adverse climatic condition have been attributed as origin for food crises in Africa [3]. Specifically, the Global Framework on Water Scarcity and Agriculture, or WASAG, operated by the Food and Agriculture Organization (FAO), highlights that water is at the nexus of agriculture and energy production, and water scarcity can further deteriorate food security and the ability to achieve the sustainable development goal of zero hunger (SDG #2) [4].

In a review article, Bhaduri et al. indicated that water is not only part of many other SDGs but in many aspects a key precondition for the goals [5]. Currently many food security programs are implemented on its own with a constrained perspective notrecognizing water issues at the local level. Nhamo et al. reviewed the availability of water resources in Southern Africa and highlighted that wise use of water resources is very critical not only in sustaining food security, but it is also critical for regional integration and economic development by increasing hydro-energy production [6]. Further, based on the case studies, Hadley and Wutich concluded that measurement of both water and food insecurity at the household level is very critical, since limited access to food and water causes psychosocial distress and mental health issues among caretakers directly affecting maternal and child nutrition status [7]. In proposing approaches to achieve SDGs, Liu et al. highlighted the importance of using nexus framework, recognizing interactions and linkages between targets and value of an integrated SDG achievement plan [8].

Water is an essential nutrient, and access to it is recognized by the United Nations as a human right. However, approximately 785 million people worldwide do not even have access to basic drinking water services, with rural areas in sub-Saharan Africa experiencing a disproportionate burden of poor water access due to low-to-no availability of formal water services [1]. Consequently, in rural and remote areas, more than half of the people rely on surface water sources, such as rivers, lakes, and wetlands, which often require long-distance walking and carrying heavy loads of water containers [1].

From the studies conducted in rural regions in sub-Saharan Africa, it has been found that limited water access is associated with quarrels, shame, and social disengagement among women [9,10,11]. Additionally, water shortage has also been associated with poor mental health, including depression, anxiety, and worry about future [9,10,11]. Further, recent literature on water insecurity scale development indicates that water insecurity increases the risk for food insecurity [12,13]. For instance, in a study by Brewis et al. in 21 low-to-middle-income countries, it was found that water insecurity was positively associated with food insecurity in urban and even rural households [12]. Similarly, in another instance, we found a dose-response relationship was demonstrated, which is, as water insecurity increased, the levels of food insecurity also increased among rural households [14]. Hence, there is evidence that limited water access is associated with food insecurity. However, it is not clear the mechanisms by which limited water access affects food insecurity at the household level. To fully understand the link, as a first step, it is important to understand water fetching and daily routine in reference to water use and how things are carried out when the availability of water is limited at the household level. Hence, with the goal of developing a framework on how limited water access is connected to food insecurity, an exploratory study using qualitative methodology was conducted in rural areas in the West region in Cameroon to (1) examine water access, its management, and its daily use; and, (2) investigate common behavior changes and coping strategies adults used in managing limited water availability in their households.

### Study Area

Cameroon, which is situated between West and Central Africa, experiences a high level of food insecurity and undernutrition issues. In 2018, the national prevalence of under-five stunted growth was 28.9%, with 16% of Cameroonians experiencing a moderate-to-severe level of food insecurity [15,16]. Cameroon has a dense network of rivers and a high annual rainfall volume, but water scarcity remains a huge issue among its inhabitants, with only 47% of the rural population having access to improved source of piped water service [17,18]. The West region is in the “French-Cameroon” and Bamileke and Bamoun are the two main tribes represented, with main religion being Christianity. The climate is favorable and characterized by high air humidity and several hundred inches of annual rainfall [19,20].

The study was specifically conducted in the Menoua Division of the West Region of Cameroon [19]. The Menoua division comprises of 22 villages, with the most populated village Bafou. It was selected for the study due to our previous established collaboration and its external validity, representing rural settings of sub-Saharan Africa, where economic water scarcity prevails more than physical water scarcity. In the Menoua Division, more than 80% of the population are farmers and the most important plants grown are coffee, tea, plantains, and Irish potatoes. Animal rearing is also widely practiced, notably pig farming [19,20].

## 2. Materials and Methods

### 2.1. Data Collection

The study was approved by the Cameroon National Committee of Ethics for Human Research and the University of North Carolina -Greensboro’s Institutional Review Board. All of the participants in the study provided written consent. The manuscript was prepared following the Standards for Reporting Qualitative Research [21]. During the study period of February to May 2019, three focus-group discussions (FGDs) and six key-informant interviews (KIIs) were conducted.

#### 2.1.1. Focus Group Discussions

Three FGDs were conducted, as shown in Table 1. FGD 1 was conducted with adult women, responsible for water fetching and management in their households. The second FGD was conducted with adult men, who reported themselves as the head of their households. Lastly, young adults aged 18 to 21 years old were invited to participate in a third FGD (Table 1). Adult men and women were recruited from villages using a door-to-door approach and snowball sampling with the help of a female local community social worker. The social worker was native to the study area, knew the water sources in villages and surrounding areas, and was fluent in the local dialect. As for the third young adult focus group, the participants were invited if heads of the household confirmed their key role in water fetching and management tasks.

All three discussions took place in a private setting on different days and were conducted in French by the primary author fluent in the language. The results of previous research conducted in the study area [22,23] and water access-related literature were used to develop the discussion guide. In alignment with the study objectives, the FGD was divided into two sections: (1) water access, its management, and daily water use; and, (2) changes in behavior and strategies used at the household level to manage limited water access. In the first section, questions were asked to establish roles and responsibilities of household members in collection and management of water use, including monitoring water use at the household level. Information on seasonal variations, i.e., dry vs. wet season, was also collected.

The discussions were audio-recorded and each of them lasted for approximately two hours. Following the key FGD principles, all of the participants were given the opportunity to respond during discussions, and as needed, prompts were used to dig deeper into issues raised during questioning.

For the analyses, the recorded discussions were transcribed word-for-word into French. Both transcripts and notes were translated into English by the primary author. Subsequently, co-authors independently reviewed and analyzed the transcripts using inductive and deductive approaches [24]. Based on the inductive content analysis approach, recurrent quotes were systematically searched across transcripts and those judged to be pertinent grouped together as subthemes or categories. Afterwards, a deductive content analysis approach was used to further look for quotes that fell within pre-defined domains of water fetching and management and change in behavior due to water scarcity. In both cases, the unit of analysis was a quote, defined as a fraction of transcript that is understandable and contains one single idea. After both of the authors conducted this phase independently, the results were compared, and disagreements were discussed until a consensus was met.

#### 2.1.2. Key Informants Interviews

The questions for KIIs were streamlined using the FGD results. The questionnaire specifically focused on in-depth investigation into the severity and occurrence of water shortage and coping changes that families made to manage and address it. During this phase, the goal was also to confirm themes and subthemes identified from FGDs and obtain significant details on them. In total, six KIIs were conducted that involved two men and four women. The study participants for the interviews were also recruited using the snowball sampling technique and door knocking approach with the help of local social worker. Mainly, participants that were local to the area and familiar with the sources of water in the area were recruited. The interviews were carried out in a private setting and lasted for 60 min on average.

The interviews were audio-taped and transcribed verbatim, and then translated from French into English for systematic analysis. The inductive and deductive approaches used to analyze FGDs were also utilized in analyzing KII transcripts. Each FGD and KII participant received an incentive of 1000 Francs CFA, roughly equivalent to US $2.

#### 2.1.3. Validation of Data was Ensured by Taking Following Multiple-Steps

Recruitment was carried out at different venues using the door-to-door approach and snowball sampling to ensure that the participants were representative of the target population of rural households in Central Africa.The triangulation strategy was used involving different data collection techniques (FGD vs. KII) and different individuals analyzing the data. The authors also used observations and results of the previous research [22,23] in the study area in order to validate the data of the current study.Additionally, deep saturation in the results was achieved after conducting FGDs and KII in a successive manner.

## 3. Results

Of the total 25 participants, 15 were female and 10 were male, with the ages ranging from 18 to 75 years, including the young adult group. About one-third of participants reported working on farms, as seen in Table 1. Among the six participants from the young adult group, five were currently in middle and high school. All of the participants, except three (one male, one female, and one young adult), reported that their households did not have access to tap water and they were involved in fetching water from one or two nearby natural sources.

### 3.1. Water Access, Management, and Daily Water Use

#### 3.1.1. Water Access

In investigating the sources of water, it was found that participants generally relied on rivers and streams for daily water use for both drinking and household chores, as shown in Table 1. For some families, water pumps or wells were accessible on a limited basis. Few households had a water pump in the compound, and sometimes allowed other villagers to use the pump. In particular, streams or ‘flowing’ water sources were considered to be pure and were preferred for drinking purposes.

#### 3.1.2. Water Fetching Routine and Responsibilities

All children and women, except for adult male members (father, grandfather), participated in water-fetching activities. Children as young as four years old were involved via carrying at least one bottle to the water source. All members engaged in water-fetching activities followed an established schedule. For instance, school-age children drew water after school, at around 4:00 p.m. Women generally used their mid-morning time to collect water.

The amount of water collected depended upon the household size, whereas household composition determined the total number of trips made to fetch water. Along with women, adolescent children were often the primary water fetchers for the households and, because they could carry large containers, the households with older children were making fewer trips than households with younger children. Additionally, the types and number of containers, the availability of trolley-like equipment, and the water output at the water source also affected the number of trips each household made to collect “adequate” water for the household. For instance, a female key informant (age: 50) said,
“Every morning, we draw water that we will be using all day and night… I had about seven cans, they got busted. I only have three cans remaining. Sometimes, when children are lazy, they just fetch those three cans. Other times, they will pour that water inside a barrel and do a second trip.”

For the majority of families, water was fetched every day. Only in certain situations, such as a child being sick or a caretaker having to attend social event, was the daily water-collection routine disrupted and, in such a case, paying a neighbor’s child in cash or kind to fetch some water was common.

#### 3.1.3. Water Access by Seasons (Wet vs. Dry)

Seasonal difference in water fetching and management was noted. In the rainy season, direct collection of rainwater in barrel containers was common and used for cleaning, laundry, and related household uses. However, it was noted that, in the rainy season, water collection from the river for drinking was problematic due to mudflows and heavy water run. For instance, one participant reported (FGD, male, age: 52),
“The water here at home is from the river. So, there is always water, but sometimes, that water can be dirty. Especially, in the rainy season, tornado carries land from the fields and all that pour into the river. When people arrive at the water source, they find that the water is dirty and they do not draw it.”

In the dry season, acute water shortage was noted, with dried-up water sources and no rainwater available for household work. Some women reported that they got up very early to get some fresh, non-muddy water for drinking purposes. As one woman said (FGD, female, age: 35),
“Water affair, especially in the dry season, it’s not easy at all. It is not easy.”

Similarly, a male participant (FGD, male, age: 22) said,
“Yes, there are times that water does not flow. Usually in the dry season, the water may finish at the water source.”

### 3.2. Behavior Changes and Coping Strategies Used to Manage Limited Water Availability

The results obtained from the FGDs and KIIs indicated that water was used in a hierarchal order for household activities: cooking was given priority, followed by hygiene with bathing and laundry. Water for sanitation was not given preference; rather, washing utensils and home gardening were prioritized over sanitation. For instance, one key informant (male, age: 55) reported,
“If the dishes are not clean, I will not even be able to cook.”

A woman (age: 42) said
“All water remaining from other activities are collected and thrown there (toilets), to at least remove bad odors from the toilets.”

Another woman from a KII (age: 50) supported that,
“If there is not enough water to clean the toilets, we wait for the next day.”

#### 3.2.1. Changing Cooking Plans

In general, households cooked in the late afternoon for the dinner and for the next day’s breakfast and lunch. Whenever water was not available, cooking and meal preparation were significantly affected. In limited water or water shortage periods, households relied on eating raw or just grilled food items, such as roasted yams, instead of a complete meal, like dry corn flour couscous and sautéed legumes. In certain case, the FGD and KII participants also reported skipping dinner, since there was no water available to cook. For instance, one participant (FGD, female, age: 34) said,
“We sleep hungry because there is no water.”

Similarly, one young adult in FGD (age 21) mentioned,
“At moments that we had water issues, we could spend a day without cooking because there was just enough water to drink for the night.”

Families were not able to cook anything in certain acute instances of complete lack of water. However, changing the meal plan and cooking something that required less water was the most common strategy among participants. Table 2 lists further quotations explaining the relationship between water availability and meal plans.

#### 3.2.2. Prioritizing Water for Drinking

As expected, in addition to cooking, water for drinking was a priority for households. When forced to choose, water for drinking was saved first. If the quantity of drinking water was insufficient, each member just drank a small amount to “refresh the throat” and waited until the next day to fetch more. Additionally, women often reported going to bed thirsty and skipping, to save drinking water for other household members. As one woman (KII, age: 50) said,
“When there is just a small amount of water, I take a small quantity only to help with ‘swallowing’ my food. Nobody can drink the amount he/she wants.”

In particular, young babies were given priority in ensuring that there was some water available for them. For households with young children, borrowing water from neighbors was a common practice. All of the participants admitted borrowing water from neighbors at least once in a while or giving water to someone asking for it. However, this strategy was not considered ideal, because it was noted that all were in the same situation. Moreover, borrowing water was generally associated with a feeling of shame of being unable to take care of the family. In an elementary school-teacher’s words, (KII, male, age: 31),
“After spending a night at home without water, I arrived in class, asked for water from my students and drank.”

Often, the source of drinking water was different from the source used to collect water for household chores. People preferred spring or flowing water for drinking purposes. However, in the dry season, when springs generally dried up or had a very slow output, the quality of drinking water was a big concern. Using brown or unclean water for drinking was common for participants. It was further noted that young children were also given water that was not clean or clear, with the idea that the sooner the children start drinking ‘local’ water, the sooner it helps them become “immune” and less likely to fall sick. Boiling water before using for drinking was reported mainly for infants, because fuel collection was another chore and wood collected for fuel was also expected to last for a certain time.

#### 3.2.3. Limiting Home Gardening to Rainy Season and Recycling Water Regularly

Home gardening was common but mainly done during rainy season. Growing crops for the household was critical, because a stable source of income was not regularly encountered among participants. Cultivating their own food was preferred, and only discretionary items, such as salt, sugar, Maggi cubes, and oil, were generally purchased. Additionally, the recycling of water was very frequent, such as water used for hand washing and laundry was saved and used to carry out other household chores, like washing compound floors. In particular, during the dry season, use and reuse of water was very habitual where water was recycled for all activities except cooking and drinking. Table 2 presents some of the quotations explaining the use and reuse of water for daily activities.

### 3.3. Link between Water and Food Insecurity, i.e., Food Production, Access, and Utilization

The amount of time spent on water fetching and its management and coping strategies people used clearly indicated that all three pillars of food security (availability, access, and utilization) were affected among participants due to limited water access, as shown in Figure 1.

Firstly, for home food production, reliance was mainly on rainwater for growing crops. It was noted that animal breeding was also affected by water availability. Growing food during rainy season was a priority to ensure sufficient supply of staple food, such as maize, beans, for the household around the year.

Secondly, access to food was significantly affected, as water fetching and its management was a huge time-cost burden, taking away time for childcare and economic activities, including educational opportunities for young adults. Women and young adults often reported that securing water was the main part of their life, excluding them from many things, such as socializing and earning extra money through work.

Thirdly, water also affected the food utilization pillar in several ways, as shown in Figure 1. One, daily intake of water was affected. Specific information on daily intake of water was not collected; however, drinking a few sips of water or feeling tired but not properly drinking water, was often noted by participants. Two, water quality was often questionable, even for participants. 

The majority of participants used open or unimproved sources of water and the use of filtering techniques was not common. Also, water for sanitation was not a priority, including hand washing and cleaning floors, due to water shortage. Recycling of water was common, increasing the risk of cross-contamination.

## 4. Discussion

This study aimed to collect information on water access and use to understand the synergistic relationship between limited water access and the three pillars of food security, i.e., availability, access, and utilization. The results of this research showed that water access was affected by the seasonality and availability of water sources, as well as household size and composition. Especially, the number of people in the adolescent age range improved water fetching, since this age group also played a primary role in drawing water along with the caretaker or mother. The presence of storage containers also determined the availability of water in the household. In investigating sources of water, as in other studies [10,25], it was discovered that a reliance on surface water, such as rivers and streams, was very common in the study area.

In estimating the potential pathways between water and food insecurity in sub-Saharan Africa, it is critical to note that agriculture is a major source of livelihood in rural households. Hence, water is directly linked to the economic status of families. In the annual report, the United Nations System Standing Committee on Nutrition indicated a strong link between water and food systems beginning with agriculture and food production. Especially, in sub-Saharan Africa, water is imperative for smallholder farmers, since food availability for them depends on their ability to grow food [26]. Further, not only food production, but the results of our study indicate that even the food processing and preparation are affected. For instance, in the study, it was found that traditional, highly nutrient-dense meals made of beans (Koki) and maize (Couscous) were replaced by other type of foods, such as grilled yams, due to lack of water. This phenomenon is very critical, since it indicates that water shortage is directly related to lack of variety in the diet and thereby food insecurity. Further investigation is warranted to fully understand how water access affects the nutritional status of households.

The results of our study showed that tradeoffs occurred for water use at the household level, such as whether water be used for drinking or for cooking or maintain a sanitary environment. Especially, water for drinking was reported as being the highest priority for water fetching and daily water use. When the availability of water was limited, household members either drank less water than they felt they should, drank unclean water, or as a last resort borrowed drinking water from a neighbor. Additionally, in a review of recently developed household-level water insecurity scales, it was found that skipping drinking or reducing water intake was one of the strongest scale items of water insecurity assessment [27]. Hence, a link between limited water access and food insecurity is also occurring by reduced daily intake of water. Water is an essential nutrient, it encompasses 75% body weight in infant to about 55 to 60% in adults [28,29]. Water keeps the body temperature normal and it is critical for food digestion and absorption at an optimal rate. By impacting water intake, limited water access is potentially affecting the third pillar of food security i.e., utilization. In the future, research should be conducted in order to understand the relationship between water insecurity and daily water intake.

The results of our study showed that water quality was also a concern. In such a case, it can mediate poor nutrition through diarrheal diseases. Water that is contaminated with pathogens can lead to diarrhea and poor food utilization, highlighting another strong link between water access and food insecurity. This is critical since diarrhea is the second leading cause of death among children [30]. In addition to direct ingestion of contaminated water, lack of access to safe and clean water, including recycling of water—using and re-using water for multiple chores—can increase the risk for infection and subsequent poor food utilization and nutritional outcomes. For instance, the use of contaminated or recycled water for cleaning or hand washing as well as conducting laundry at the water source can increase the risk for schistosomiasis infection via skin penetration. Hence, even when direct ingestion does not occur, lack of access to clean water can increase risk for infection due to uncleanliness in the surroundings. As demonstrated by the MAL-ED study, poor environmental conditions generate chronic intestinal inflammation and poor absorptive function, and thereby stunted growth and reduced cognitive development among children [31].

In addition to affecting food availability and utilization pillars of food security, water also potentially affects food access. Mainly, as indicated in the results, women played a key role in water fetching and its daily management. In such a case, limited water access increases time cost that women spent for water fetching, affecting their ability to care of themselves and their children. Additionally, due to time cost for water fetching, women do not get a chance to engage in income generation activities and their ability to improve food affordability and access for the households.

Our study used a strong methodological approach of combining both FGDs and multiple KIIs to capture important themes that are related to poor water access. This allowed us to demonstrate different pathways through which three pillars of food security (availability, access, and utilization) are affected by limited water access. However, the results of this study can only be generalizable to other rural settings with similar types of water sources. In the future, longitudinal, quantitative study measuring daily water use per capita, meal patterns, and hydration status in adults and children are warranted in order to assess and quantify short and long-term impact of limited water access on food insecurity and nutritional status.

## 5. Conclusions

In conclusion, communities from rural areas in sub-Saharan Africa experience limited access to water that is characterized by walking long distances to the water sources and taking multiple trips daily to collect adequate amounts of water for their households. This paper highlights the key water-food linkages emphasizing that the Sustainable Development Goal (SDG) #2 of ending hunger and SDG #6 ensuring availability and sustainable management of water and sanitation for all are highly connected. The targets in SDG #2 cannot be achieved unless advancement is made in SDG #6. Hence, efforts recognizing intersectionality between water and food security and putting priority for water-sensitive nutrition interventions are critical. Furthermore, focusing both on quantity and quality of water using context-specific approach will help to achieve several SDGs. In the rural setting, besides distance to water source, the household size, household composition (age repartition of family members), and the number and availability of water storage containers also affect water access. Research initiatives further investigating how intra-familial factors affect water availability at the household level are warranted. In sum, government, policy makers, and health organizations should recognize the link between water and food security and design collaborative projects to develop sustainable and effective health and nutrition programs. Efforts are also needed to join forces with the water managers from the household to community to the regional level, in order to strengthen the pathways between water and food not only at the micro, but also at the macro levels.

## Figures and Tables

**Figure 1 ijerph-17-04848-f001:**
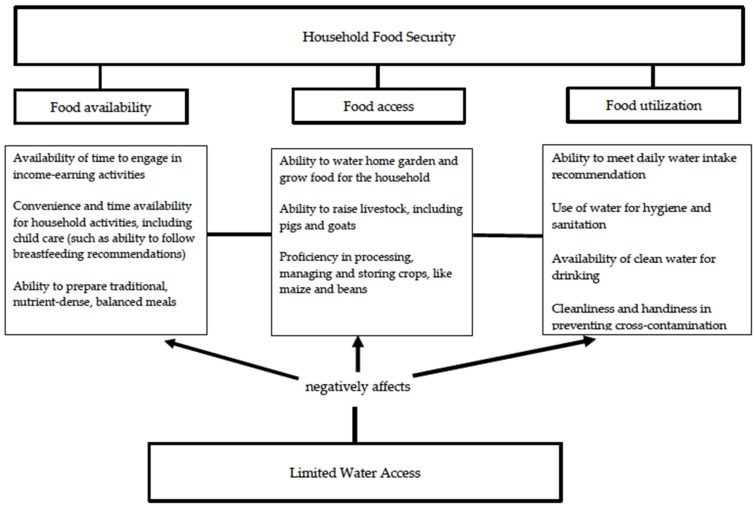
Diagram of potential pathways through which limited water access increases the risk of food insecurity among families living in the West Region in Cameroon.

**Table 1 ijerph-17-04848-t001:** Socio-Demographic Characteristics of Focus Group Discussion and Key Informant Interview Participants in the West Region in Cameroon (n = 25).

Description of Participantse	n
FGD group 1 (only women, >21 years old)	8
FGD group 2 (only men, >21 years old)	5
FGD group 3 (mixed18 to 21 years old) ^†^	6
KII (mixed, >21 years old) ^‡^	6
Socio-Demographic Characteristics (n = 25)	n (%) ^¶^
Gender	
Female	15 (60)
Male	10 (40)
Education ^§^	
Lower than high school	18 (72)
High school or above	7 (18)
Occupation	
Small-scale food/grocery seller/service providers	9 (36)
Working on farms	8 (32)
Salaried	2 (8)
Students	6 (24)
Common water sources	
Taps/borehole/water pumps Wells Rivers/streams	6 (24) 2 (8) 17 (68)

FGD: Focus group discussion; KII: Key informant interviewee; ^†^ mixed young adult group of 18 to 21 years old sons and daughters living with their parents, three males and three females; ^‡^ two men and four women; ^§^ it also includes young adults who were still pursuing their education and do not represent their final education level. ^¶^ Percentages are rounded to the nearest full digit.

**Table 2 ijerph-17-04848-t002:** Examples of Quotes Related to Changes in Meal Plan and Use of Water in Multiple Purposes to Manage Limited Water Among Rural Households in the West Region in Cameroon.

Changes in Meals and Food Cooked in Limited Water Availability
“If you have one can (20 L)? Couscous! You directly abandon the idea of cooking it (laughing). When there is not enough water, I change the menu, I only cook sautéed rice.” FGD, women, age: 35, HS = 8
“I was planning to cook tomato stew and rice. The rice itself, half a kilo, takes a certain amount of water. Being able to wash the tomatoes with all the condiments demands a certain amount of water. I see there is not enough, and it is better to cook sautéed pasta. For pasta, 2 L will be a lot to cook.” KII, man, age: 55, HS = 1
“I can’t cook Koki with 5 L of water. It won’t be enough even to wash Koki grains (black-eyed peas); I will just change the meal plan. Or if I just have Koki as food available, I go to the river to fetch for water.” KII, woman, age: 41, HS = 6
“Mashed potatoes and beans do not take a lot of water.” KII, woman, age:42, HS = 9
**Use of Water for Multiple Purposes or Recycling**
“With water for the laundry, you clean the toilets to avoid odor.” KII, woman, age: 42, HS = 9 “I removed solid particles inside water that I cleaned food items with (rice etc.) and I water the floor with the rest.” KII, woman, age: 42, HS = 9 “If I wash baby clothes, I can use the remaining water to wash her brothers’ clothes with it.” KII, woman, age 31, HS = 7 “The rest of water I cook with to wash plates …and the remaining water for the dishes to water the floor.” KII, woman, age 31, HS = 7 “Water we washed the dishes with can be used it to wash feet, wash floor, as well as toilets. Water remaining after taking a bath, we use to wash dirty coats used for farming.” KII, man, age: 31, HS = 5 “The water I used to wash clothes, I can re-use that water to wash floor, even toilets.” KII, woman, age: 41, HS = 6 “The water we already used to wash clothes, or dishes, we water floor with it.” FGD, man, age: 58, HS = 3

FGD: focus group discussion; KII: key informant interview; HS: household size. These quotes are from 25 men and women who participated in focus groups or in key informant interviews, age ranging from 18 to 75 years.

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
