# Peer review of "A Qualitative Examination of Water Access and Related Coping Behaviors to Understand Its Link to Food Insecurity among Rural Households in the West Region in Cameroon"

_ijerph, 2020, doi:10.3390/ijerph17134848_

Round 1

Reviewer 1 Report

While this manuscript has relevant findings in terms of yielding a better qualitative understanding of water insecurity in Cameroon, the framing of the main research questions is misleading and results do not adequately support the conclusions presented. The title of the paper leads the reader to think that an association will be demonstrated between food and water security through the analysis, while the abstract and introduction indicate that child and maternal undernutrition will be a core topic addressed. Yet, all the data (except the quotes presented in section 3.2.1 Changing Cooking Plans) are focused on issues related to water insecurity and the connection to food insecurity is made primarily on a theoretical model (Figure 1.0). The issue of maternal/child undernutrition is likewise unaddressed with data. While making connections between water and food insecurity would be appropriate for a Discussion section, the current connection with food insecurity is not robust enough (as there is not much qualitative, or any quantitative, data presented) to warrant the current framing or conclusions.  

Author Response

Thanks for your feedback. Attached is the response to comments document.

Reviewer 2 Report

Good abstract but it need to be developped by adding one or two sentences on the how the authors see the solution by proposing differents way towards the managers and decision makers

No Context is proposed ?

Methodology of datation validation and accuracy of data

Study Area have to be outside of methodology part

Map of study area is missed

Synthesis table of population sample

The study is based on a few population : where the representativity of this study

No think on the sampling methodology

There is a mix between economic situation traduced by this sample of population and accessibility is little bit disturbed by this choice way

Conclusion is so short and don’t give any perspectives of proposal to reduce this scarcity by developing indicators.

Author Response

Thanks for the feedback. Please find attached response to comments document.

Reviewer 3 Report

The topics is of the highest public health importance.

To get information on the problem and the solutions from those affected and involved is not usual and critical.

i suggest the following suggestions for improving the publication.

1/ On the methodology, how people have been selected?Any bias? Could the physical, social and economic environment of the communities be a little bit described.

2/ On the description of the interactive synergy of water/ food insecurity, could you develop any high risk situation identified?( for exemple, pre- harvest season, absenteism, external factors..)

3/ On the coping strategies,they seem to concentrate on water and food utilization. What about availability and access? Any income generating activities for better water/ food security? 

4) On discussions, could you indicate some recommendations for local policy changes, health services and community initiatives?

Author Response

(The authors gave the same response as above.)

Reviewer 4 Report

- Line 29. Please remove the dot before "1".

- It should be given information on what percentage of the region’s population took part in the FGDs.

- Line 86-98. The presentation of FGDs should be given in table. It would be more readable.

- In the test, please mark the statements of people participating in discussions in quotes. Reading the manuscript I had a problem when such a person's speech ends. You can give quotation marks and italics. For example:

Lines 160-163. For instance, a female key informant (age: 50) said,

Every morning, we draw water that we will be using all day and night... I had about seven cans, they got busted. I only have three cans remaining. Sometimes, when children are lazy, they just fetch those three cans. Other times, they will pour that water inside a barrel and do a second trip.

-Line 208-209. Please remove a dot at the end of the heading of Table 2. (it is doubled).

-Line 253. This sentence “To begin w” is not complete. The rest is in line 257 (under the Figure 1). Please correct this.

-Please change the diagram on Figure 1. It is not readable for Reader. In the second columns there is unfinished sentence.

-It could be interesting for Reader to know how far was the walking distance for water (for participants in the research) and how long does it take. This would emphasize the seriousness of the problem.

Author Response

(The authors gave the same response as above.)

Round 2

Reviewer 1 Report

Dear Authors,

 Thank you for the response and edits. However, the 'response to revisions' is quite vague and reading through the paper again... the same problem remains... the bulk of the data presented is focused on water insecurity.

I understand that the idea is to use this evidence to hypothesize particular pathways linking water and food insecurity, but the hybrid combo of data/theory is still not convincing. A more effective approach might be to draw from literature that has already been published on the water-food nexus (Webb, 2006; Hadley and Wutich, 2009; Conway et al., 2015) to identify pathways connecting the two, then bring in examples from your study that support these hypothesized pathways.

Unfortunately, even with the revisions, the manuscript still feels disconnected and misleading. If it was a "qualitative examination of water access to understand its link to food insecurity" - why weren't there any questions about food insecurity? I think the authors are trying to do more with the data they have, which is a good thing, but as mentioned in my first review, the framing of the paper is still not appropriate. 
